# DS-CNN: Deep Convolutional Neural Networks for Facial Emotion Detection in Children with Down Syndrome during Dolphin-Assisted Therapy

**DOI:** 10.3390/healthcare11162295

**Published:** 2023-08-14

**Authors:** Jesús Jaime Moreno Escobar, Oswaldo Morales Matamoros, Erika Yolanda Aguilar del Villar, Hugo Quintana Espinosa, Liliana Chanona Hernández

**Affiliations:** Escuela Superior de Ingeniería Mecánica y Eléctrica, Unidad Zacatenco, Instituto Politécnico Nacional, Ciudad de México 07340, Mexico

**Keywords:** Down Syndrome, deep convolutional neural network, deep learning, dolphin-assisted therapy, facial emotion detection

## Abstract

In Mexico, according to data from the General Directorate of Health Information (2018), there is an annual incidence of 689 newborns with Trisomy 21, well-known as Down Syndrome. Worldwide, this incidence is estimated between 1 in every 1000 newborns, approximately. That is why this work focuses on the detection and analysis of facial emotions in children with Down Syndrome in order to predict their emotions throughout a dolphin-assisted therapy. In this work, two databases are used: Exploratory Data Analysis, with a total of 20,214 images, and the Down’s Syndrome Dataset database, with 1445 images for training, validation, and testing of the neural network models. The construction of two architectures based on a Deep Convolutional Neural Network manages an efficiency of 79%, when these architectures are tested with a large reference image database. Then, the architecture that achieves better results is trained, validated, and tested in a small-image database with the facial emotions of children with Down Syndrome, obtaining an efficiency of 72%. However, this increases by 9% when the brain activity of the child is included in the training, resulting in an average precision of 81%. Using electroencephalogram (EEG) signals in a Convolutional Neural Network (CNN) along with the Down’s Syndrome Dataset (DSDS) has promising advantages in the field of brain–computer interfaces. EEG provides direct access to the electrical activity of the brain, allowing for real-time monitoring and analysis of cognitive states. Integrating EEG signals into a CNN architecture can enhance learning and decision-making capabilities. It is important to note that this work has the primary objective of addressing a doubly vulnerable population, as these children also have a disability.

## 1. Introduction

An incidence of Trisomy 21 is estimated in the world to be between 1 in 1000 and 1100 newborns. In Mexico, according to data from the General Directorate of Health Information (2018), there is an annual incidence of 689 newborns with Trisomy 21 [1]. Zhao et al. in [2], mention that Trisomy 21, known in society as Down Syndrome, is a disease of genetic origin characterized by the presence of a third chromosome, in which it should be the 21st pair. The purpose of the current study (Section 2) is to use artificial intelligence in favor of certain medical conditions through therapies that can be considered regular. It is important to mention that patients with various pathologies not only resort to regular therapies but also to alternative therapies, either herbal or assisted by animals, such as horses, dogs, and even dolphins. One of these alternative therapies used in children with Down Syndrome is dolphin-assisted therapy, which is focused on reducing anxiety and stress levels, as well as physical improvement. Dolphin-assisted therapy has been developed as an alternative treatment for people with various physical disabilities and psychological disorders, such as Autism, Attention Deficit, Trisomy 21, Spastic Cerebral Palsy, and Obsessive Compulsive Disorder. This therapy is aimed at complementing and reinforcing existing therapies but does not replace them [3]. Being a therapy aimed at people who have difficulties to communicate in general, it is difficult to determine its level of effectiveness.

Emotions are part of human life, and although sometimes we try to hide them, they occur involuntarily; they are reflected in microexpressions and macroexpressions. These facial expressions are very useful for a person with Trisomy 21 to communicate better, since they mostly use nonverbal communication [4]. Facial emotions (FEs) are identified by being either voluntary or nonvoluntary. Voluntary FEs can last between 0.5 to 4 s, and they also cover a large facial area that can be perceived in the entire area of the image with very notable gestures, while the nonvoluntary FEs are characterized by having small muscular movements with a very short duration, between 0.065 to 0.5 s, in a natural way in a small facial area, demonstrating the emotions that are trying to be hidden [5].

In addition, artificial vision has become a great tool to develop a recognition algorithm, as it combines five tools from computer science: (i) computer science concepts, (ii) digital image processing techniques and ideas, (iii) recognition of patterns, (iv) artificial intelligence, and (v) computer graphics. Most machine vision tasks are related to the process of obtaining information about events or descriptions from input scenes based on four main features:Human vision;Pattern recognition;Computer science;Signal processing.

The knowledge of the emotions of children with Down Syndrome (DS) obtained through the analysis of their facial expressions during an assisted therapy with dolphins using Artificial Vision and Deep Convolutional Neural Networks can significantly contribute to the effectiveness of the therapy. Understanding the emotional responses of DS children during therapy sessions can provide valuable insights into their level of engagement, comfort, and overall well-being. This information can help therapists and caregivers tailor the therapy sessions to meet the specific emotional needs of each child, enhancing their overall experience and potentially improving therapeutic outcomes. Additionally, monitoring and tracking emotions throughout therapy can aid in identifying patterns or triggers that may impact the child’s progress, allowing for timely adjustments in the treatment plan. Ultimately, having a deeper understanding of the emotions of DS children during assisted therapy with dolphins can foster a more supportive and empathetic therapeutic environment, promoting their emotional development and overall growth.

The objective of this research is to develop a facial expression recognizer for children with DS using a deep learning neural network. This paper does not explicitly mention the number of methodologies evaluated, but it focuses on using a deep learning approach to achieve the objective. Planning the proposal by contrasting related works means comparing and analyzing existing research and methodologies in the field of facial expression recognition for children with DS to identify gaps and potential improvements in the proposed approach. This work provides a theoretical understanding of concepts by explaining the neural network architecture and the principles of deep learning. In order to achieve this objective, the following specific objectives must also be met:Evaluate the methodologies focused on analyzing emotional responses for the planning of the proposal by contrasting related works.Establish a theoretical understanding of the necessary concepts to design an alternative in emotion identification through facial emotion detection.Define the processes and threads necessary to determine facial emotion detection.Analyze the results originated during the experimentation to identify the variables that modify the efficiency of facial emotion detection.

To understand the neural responses and changes that occur during dolphin-assisted therapy (DAT) sessions, some state-of-the-art works measure both the dolphins’ and children’s brain activity. In this way, Moreno Escobar et al. in [3], and Matamoros et al. in [6], focus their works on measuring and analyzing both dolphin and child brain activity, respectively, during DAT to identify significant differences in neuronal signals. The researchers use electroencephalographs to record electrical brain activity by placing electrodes on the scalps of both dolphins and children. This allows them to monitor and analyze changes in the power spectrum density of brain activity during DAT, as shown in Figure 1. The relevance of measuring dolphins’ brain activity lies in gaining insights into the potential effects of DAT on brain responses, helping to explore the therapeutic benefits and mechanisms of the therapy for children with disabilities. However, it is essential to consider other factors, such as the emotional state of the patients, to ensure that the observed brain changes are indeed attributed to the therapy and not influenced by other external factors. In this way, Figure 1 also illustrates how a female adult dolphin engaged in DAT remains attentive, as indicated by the prevalence of high frequencies (15–30 Hz), while in children, there is an increase in low frequencies (0.5–8 Hz), which are predominant for their age. Figure 1b presents the brain activity of a child before (in red), during (in blue), and after (in green) participating in a DAT, clearly indicating enhanced cerebral activity during the therapy. This increment could arise from various factors, such as the mere presence of the dolphin. Consequently, these increments might be influenced by the emotions experienced during DAT. Hence, it is crucial to selectively subsample moments from EEG time series when the child is at ease and eliminate potential emotional biases to ensure accurate analysis to subsequently correlate them with the corresponding moments of dolphin brain activity and estimate the strength of the relationship between the two electroencephalograms.

In this way, one possible solution for the main problem is to analyze the facial expressions of children with Down Syndrome to identify emotions during an assisted therapy with dolphins through artificial vision by means of a Deep Convolutional Neural Network, since the identification of feelings during DAT by means of this deep learning tool is the first step to know the benefits of these kinds of therapies. These emotions are part of human life, and although sometimes we try to hide them, they occur involuntarily; they are reflected in microexpressions and macroexpressions. These facial expressions are very useful for a person with Trisomy 21 to communicate better, since they mostly use nonverbal communication [4]. Furthermore, facial emotions (FEs) are identified as being either voluntary or nonvoluntary. Voluntary FEs can last between 0.5 to 4 s, and they also cover a large facial area that can be perceived in the entire area of the image with very notable gestures, while nonvoluntary FEs are characterized by having small muscular movements with a very short duration, between 0.065 to 0.5 s in a natural way in a small facial area, demonstrating the emotions that are trying to be hidden [5].

There are several techniques to be explored for improving performance, including the following:Deep learning (DL) is a part of machine learning (ML), where algorithms such as neural networks are used to obtain data for learning in the form of system feedback. Thus, DL performs tasks repeatedly and increases the efficiency of its results to reduce learning errors. Also, DL is used for more advanced image analysis in prediction, leading to increased applications in research.Convolutional neural networks (CNNs) make use of images that are divided into labeled pixels. These labels are used to perform convolutions, which is a mathematical process in which a third function is generated from two functions for a prediction about what is being seen. As these convolutions are performed, the predictions are verified to recognize the images.Artificial vision, or machine vision, is part of science and technology, which is directed to the creation and improvement of computers to obtain greater speed and autonomy. One of the main objectives is to provide computers with sensory capacity so that they can interact with their environment like a human; this is carried out through the use of sensors, noise filtering, and development board programming.

In this paper, we present a comprehensive study on facial recognition in children with Down Syndrome (DS) using a deep neural network (DNN) combined with EEG signals and a set of images. This work is structured into four main sections. In Section 2, we provide a detailed overview of related work, discussing prior research and approaches in the field. Section 3 outlines the methodology employed for facial recognition, focusing on the integration of electroencephalogram (EEG) signals and images in the DNN model. In Section 4, we present the results of our experiments and analyses, providing insights into the performance and effectiveness of the proposed approach. Finally, in Section 5, we draw general conclusions, summarizing the findings and highlighting the potential implications of our research in the context of improving facial recognition techniques for children with DS.

## 2. Related Work

Nowadays, there is a wide variety of therapies that favor or promise to contribute to a much fuller development of children with Down Syndrome (DS). These range from the most conventional, such as physiotherapy, occupational, and behavioral therapy or speech therapy, which seek to correct oral and written communication disorders, mentioned by *Esther Martín* at [7], to alternative therapies that turn out to be of more of a complementary nature to the conventional ones. Within the latter are therapies with nutritional supplements, drugs, cells, or with animals, there are even therapies that seek healing through rites of faith and prayer, all of which are described by Roizen in [8]. This work is focused in depth on dolphin-assisted therapies. Nowadays, there is a diverse range of therapies available for children with DS, including not only the above-mentioned ones but also speech therapy, occupational therapy, physical therapy, and cognitive behavioral therapy. These therapies aim to address various developmental challenges and enhance the overall well-being of children with DS. Regarding the integration of artificial intelligence (AI) in these therapies, recent advancements have shown promising potential. AI technologies, such as machine learning algorithms and natural language processing, are being used to develop personalized therapy programs tailored to the specific needs and abilities of each child with DS. AI-based tools can assist in speech and language assessment, provide real-time feedback during therapy sessions, and facilitate interactive learning experiences. These innovative approaches have the potential to improve the effectiveness, efficiency, and accessibility of therapies for children with DS, ultimately promoting their development and quality of life. By discussing the different types of therapies used for children with DS and highlighting the role of AI in these therapies, we aim to provide a comprehensive understanding of the evolving landscape of therapeutic interventions for this population.

Sampathila et al. in [9], propose an intelligent deep learning algorithm for identifying white blood cells that generate the overproduction of lymphocytes in the human body’s bone marrow, leading to the development of acute lymphoblastic leukemia. They aimed to detect this rare type of blood cancer in children at a very early stage due to the possibility of patients being cured when this deadly disease is detected in time. The proposed algorithm uses microscopic blood smear images as input data and is implemented using a convolutional neural network to predict which blood cells are leukemia cells. The customized ALLNET model was trained and tested from available microscopic images as open-source data, achieving 96% accuracy, 96% specificity, 96% sensitivity, and a 95% F1-Score for this classifier. This intelligent method can be applied during prescreening to detect leukemia cells during complete blood count and peripheral blood analysis.

Krishnadas et al. in [10], applied YOLOv5 and YOLOv4 scaled models for automated detection and classification of types of malaria parasites and their progression stage in order to generate a faster and more accurate diagnosis for patients; the authors use a set of microscopic images of parasitized red blood cells, as well as another set of images to train the model. The YOLOv4 model had 83% accuracy and the YOLOv5 model achieved 79% accuracy. Both models can support doctors in a more accurate diagnosis of malaria and in predicting its stage. Chadaga et al. in [11], conduct a systematic literature review regarding artificial intelligence (AI) models for accurate and early diagnosis of monkeypox (Mpox), selecting 34 studies with the following thematic categories: Mpox diagnostic tests, epidemiological modeling of infection spread by Mpox, drug and vaccine discovery, and media risk management. The authors explained the detection of Mpox using AI and categorized machine learning and deep learning applications to mitigate Mpox. And Chadaga et al. in [12] developed a decision support system using machine learning and deep learning techniques, such as deep neural networks and one-dimensional convolutional networks, to predict a patient’s COVID-19 diagnosis by applying clinical, demographic, and blood markers, supporting the results of the standard RT-PCR test. Additionally, the authors applied explainable artificial techniques such as Shapley additive values, ELI5, interpretable local model explainer, and Qlattice, resulting in a stacked multilevel model with 94% accuracy, 95% recall, a 94% F1-Score, and a 98% AUC. The models proposed by these authors can be used as a decision support system for the initial detection of coronavirus patients, streamlining medical infrastructure.

Fraiwan et al. in [13], evaluates human emotions with significant benefits in numerous domains, including medicine. A machine learning model is developed in order to estimate the level of enjoyment and visual interest experienced by individuals while engaging with museum content, for instance. The model takes input from 8-channel electroencephalogram signals, which are processed using multiscale entropy analysis to extract three features: mean, slope of the curve, and complexity index (i.e., area under the curve). In addition, this scheme reduces the number of features using principal component analysis without a noticeable loss of precision. In this research, the potential of leveraging electroencephalogram signals and machine learning techniques to accurately estimate human enjoyment and visual interest is demonstrated, thereby contributing to the advancement of emotion evaluation in various fields. Shehu et al. in [14], employ a residual neural network (ResNet) model to propose an adversarial attack-resistant approach for analyzing emotion in facial images using landmarks, specifically addressing the detection of small changes in the input image. Their findings demonstrate a decrease of up to 22% in vulnerability to attacks and significantly reduced execution time compared with the ResNet model. Ngoc et al. in [15], propose a graph convolutional neural network that utilizes landmark features for facial emotion recognition, called a directed graph neural network. Nodes in the graph structure were defined by landmarks, and the edges in the directed graph were built to capture emotional information through the inherent properties of faces, such as geometric and temporal information. To address the vanishing gradient problem, the authors employed a stable form of temporal block in the graph framework, and their approach proved effective when fused with a conventional video-based method. The proposed method achieved advanced performance on the CK+ and AFEW datasets, with accuracies of 98.47% and 50.65%, respectively. Chowdary et al. in [16], use pretrained networks, including ResNet50, VGG19, Inception V3, and MobileNet, to recognize emotions using facial expressions in order to address the problem. To achieve this, the authors remove the fully connected layers of the pretrained ConvNets and add their own fully connected layers, which are suitable for the number of instructions to be performed. The newly added layers can only be trained to update the weights, resulting in an accuracy of 96% for the VGG19 model, 97.7% for the ResNet50 model, 98.5% for the Inception V3 model, and 94. 2% for the MobileNet model. Kansizoglou et al. in [17], generate two Deep Convolutional Neural Network models to classify emotions online, using audio and video modalities for responsive prediction when the system has sufficient confidence. The authors cascade a long short-term memory layer and a reinforcement learning agent, which monitors the speaker, to carry out the final prediction. However, each learning architecture is susceptible to highly unbalanced training samples, so the authors suggest the usage of more data for highly misclassified emotions, such as fear. Finally, Kansizoglou et al. in [18], develop a human–robot interaction system, where a robot gradually maps and learns the personality of a human by conceiving and tracking individual emotional variations throughout their interaction. Facial landmarks of the subject are extracted, and a suitably designed deep recurrent neural network architecture is used to train the model. The authors found that this architecture significantly aids in estimation performance, thereby benefiting the system’s efficiency in creating an accurate behavioral model of the speaker.

## 3. Methodology

The proposal of an alternative method to existing ones is divided into three main parts:
Tuning a convolutional neural network across exploratory data analysis;Adapting findings in a Down Syndrome Dataset;Training the convolutional neural network through electroencephalogram.

### 3.1. Tuning a Convolutional Neural Network across Exploratory Data Analysis

Figure 2 shows the general scheme of our proposal. Thus, this work is based on labeling the emotion a child with Down Syndrome has during dolphin-assisted therapy with high probability. In this figure, it can be observed that emotions are classified by a convolutional neural network model which predicts the possible emotion of the child during the DAT.

The use of exploratory data analysis (EDA) with three folders typically refers to the practice of dividing the dataset into three separate subsets: training, validation, and testing. This partitioning allows for a comprehensive examination of the data and helps in understanding its main characteristics and patterns. The training set is used to teach the deep learning neural network, enabling it to learn from the data. The validation set is employed to fine-tune the model and make adjustments to optimize its performance. Finally, the testing set serves as an independent dataset to assess the model’s generalization and evaluate its accuracy on unseen data. Having these three folders in the EDA process facilitates a robust and thorough analysis of the dataset, ensuring the neural network’s effectiveness and reliability in real-world applications.

In this way, this project uses two datasets, as shown in Figure 3. The exploratory data analysis is divided into test, train, and validation. The size of the train dataset is 15,109 images, the test dataset is defined by 128 images, and the validation dataset has 4977 images. Figure 3a shows a sample of this dataset. Also, in Figure 3b, the size and distribution of EDA is depicted. This approach involves performing EDA and creating initial models from scratch. During EDA, properties of the input images, classes, and data distribution are examined. The different classes of faces and their characteristics are as follows: happy faces have a smile and eyes somewhat closed with visible teeth, sad faces have eyebrows furrowed and a mouth with the opposite shape of a happy face, neutral faces have no particular expression and share some features with sad faces, and surprised faces have open mouths and eyes wide open, often with hands on the face in many samples.

The testing dataset (128) is smaller than the validation dataset (4977), because the testing dataset serves a different purpose in the model evaluation process. The validation dataset is used for fine-tuning the deep learning neural network and making adjustments to optimize its performance during the training process. It helps in selecting the best model based on its performance on the validation data. On the other hand, the testing dataset is used as an independent dataset to assess the model’s generalization and evaluate its accuracy on unseen data. It is crucial to have a smaller testing dataset to ensure that the model’s performance is not overfitted to a specific subset of data. Using a smaller testing dataset helps in obtaining a more reliable and unbiased estimation of the model’s true performance on new, unseen data. Additionally, the reference to early stopping in the original response suggests that the smaller validation dataset may be used in conjunction with early stopping techniques. Early stopping is a regularization technique that monitors the model’s performance on the validation dataset during training. If the model’s performance on the validation data does not improve or starts to degrade, the training process is halted early to prevent overfitting. This practice is commonly used to avoid training for too many epochs and to find the optimal point at which the model’s generalization is at its best.

In this case, the original training dataset probably had 128 samples, and after applying data augmentation techniques, the dataset expanded to 4977 samples. This augmentation process enhances the neural network’s ability to learn and generalize patterns from the expanded dataset, leading to improved model performance on unseen data during the training process. Furthermore, it addresses the issue of dataset imbalance. After data augmentation, the data distribution in the training dataset increased to 4977 samples, which is significantly larger than the original dataset. Data augmentation is a technique used to artificially increase the size of the training dataset by applying various transformations to existing samples, such as rotations, translations, flips, and changes in brightness and contrast. These transformations create new samples that are variations in the original data, introducing diversity and reducing the risk of overfitting.

Although the class definitions are clear, there are some images that are difficult to classify, such as happy faces that are not smiling and sad faces that are closer to being neutral. This can cause confusion for the model during classification. Additionally, there is an imbalance in the distribution of classes, with fewer images for the surprise class in the training set. This could result in a biased model that struggles to classify surprise faces.

The dataloader used is based on ImageDataGenerator class, which provides images in batches and performs data augmentation through flips, brightness adjustments, and shear operations to increase the number of images and prevent overfitting. Data augmentation is applied to the training, validation, and test sets.

Initially, there was some uncertainty about the appropriate color mode for the images, but it was determined that greyscale was suitable, since the images were found to be in greyscale using rasterio. However, the first model was constructed using three channels. Greyscale images are often considered better than color images for certain tasks, especially when dealing with limited computational resources and specific image analysis tasks. One of the main advantages of using greyscale images is their reduced data size and complexity. Greyscale images contain only one channel of pixel values (usually representing intensity), whereas color images have multiple channels (e.g., RGB channels) that require more memory and computational resources to process. For tasks that mainly rely on texture or intensity information, such as edge detection, image segmentation, and certain feature extraction tasks, greyscale images can be sufficient and more computationally efficient. Since color information may not always be relevant for such tasks, using greyscale images allows the model to focus on the relevant features and reduce unnecessary noise from color variations. Moreover, in the context of convolutional neural networks (CNNs), using greyscale images can reduce the number of parameters and, thus, the complexity of the model, leading to faster training and inference times. Greyscale CNNs are particularly useful when working with small datasets or resource-constrained environments. However, it is essential to note that the choice between greyscale and color images depends on the specific task and the nature of the data. For tasks that heavily rely on color information, such as object recognition or image classification, color images may provide more relevant features and lead to better performance. The decision to use greyscale or color images should be based on the requirements of the application and the specific characteristics of the dataset [19].

It is a CNN model with three convolutional blocks and two fully connected layers, with 605,572 parameters, and it was trained for 20 epochs with a batch size of 32 using the Adam optimizer, with a learning rate of 0.001 and categorical cross-entropy as the loss function.

In this way, the final implementation is proposed to implement a Deep Convolutional Neural Network, taking into account the advantages of these kind of architectures. The architectural design of the proposed implementation involves a Deep Convolutional Neural Network (DCNN), leveraging the advantages of such architectures. This DCNN model is structured with various layers, including convolutional, batch normalization, max pooling, and dropout layers. In this way, the presented architecture consists of 6 convolutional layers with *ELU* activation, followed always by a batch normalization layer. The first two layers have a kernel size of (5,5), while the remaining layers have a kernel size of (3,3). Another distinction is that the first two layers have 64 filters, which are doubled every two layers to reach a total of 256 filters. The model is compiled using the Adam optimizer with a learning rate of 0.001, utilizing categorical cross-entropy as the loss function and accuracy as the metric. This architecture’s goal is to effectively classify data into various classes using the CNN-based approach.

Thus, this DCNN model consists of a total of 2,398,404 parameters. Out of these, 2,396,356 parameters are trainable, meaning they are updated during the training process to optimize the model’s performance. The remaining 2048 parameters are nontrainable, indicating that their values are fixed and do not change during training. These nontrainable parameters might include biases or other fixed elements in the model’s architecture. The combination of trainable and nontrainable parameters contributes to the overall complexity and functionality of the CNN model.

### 3.2. Adapting Findings in a Down Syndrome Dataset

In Figure 4, the Second or Down Syndrome Dataset (DSDS) consisting of three folders, i.e., test, train, and validation, is shown.

DSDS also consists in three subsets: (i) *test*, (ii) *train*, and (iii) *validation*. It is important to mention that the images in the DSDS were obtained before, during, and after a female patient diagnosed with Down Syndrome participated in dolphin-assisted therapy. These images were labeled by a psychology expert. The inclusion of each image in the training, validation, or testing category was decided randomly by the same expert. In this way, the size of the Train dataset is 933 images, the Test dataset is defined by 128 images, and the Validation dataset has 384 images. Figure 4a shows a sample of this dataset. Figure 4b depicts the size and distribution of the DSDS. Each folder in this dataset has four subfolders:attention: Images of children with Down Syndrome who were in attention during a DAT.calm: Images of children with Down Syndrome with calm facial expressions.dislike: Images of children with Down Syndrome with upset facial expressions.surprise: Images of children with Down Syndrome who have shocked or surprised facial expressions.

As we can see, the DSDS is smaller than EDA because it is the best architecture found in EDA, and then it is applied in the Down Syndrome Dataset. Figure 5 shows the QR code in Google Drive, where both the EDA and DSDS can be downloaded.

The performances of the different models on the validation and testing sets are represented in Table 1. As can be seen from the results, the complex neural network architecture achieved the best performance on the validation set, and this complemented the robustness performance, achieving 75% for this image database, as shown in Figure 6. The VGG16 architecture achieved the second best performance in the validation set with 56%. Finally, EfficientNet and ResNet V2 had the worst robustness accuracy among all the models with only 25% accuracy. In this case, we employ robustness accuracy to refer to the ability of a model or system to maintain high accuracy and performance even when facing challenges, variations, or perturbations in the data or input conditions. In the context of facial emotion recognition, or any other machine learning task, a robust model can handle different lighting conditions, facial expressions, camera angles, and other variations in the input data without significantly compromising its accuracy. A robust model is less sensitive to small changes or noise in the input, making it more reliable and suitable for real-world applications where the data may not always be consistent or perfectly controlled. Evaluating a model’s robustness accuracy helps ensure that it can perform well in various practical scenarios and generalizes effectively to unseen data. Achieving high robustness accuracy is crucial for deploying machine learning systems in real-world settings where variability and uncertainties are inherent.

Thw VGG16, ResNet V2, and EfficientNet neural network models have the following architecture: The first convolutional layer has 24 filters, a kernel size of 3 × 3, and the same padding. The input shape is (32,32,1), with a LeakyRelu layer with a slope of 0.1. The second convolutional layer has 32 filters, a kernel size of 3 × 3, and the same padding, with a LeakyRelu layer with a slope of 0.1, a max pooling layer with a pool size of 2 × 2, and a batch normalization layer. The third convolutional layer has 32 filters, a kernel size of 3 × 3, and the same padding, with a LeakyRelu layer with a slope of 0.1. The fourth convolutional layer has 64 filters, a kernel size of 3 × 3, and the same padding, with a LeakyRelu layer with a slope of 0.1, a max pooling layer with a pool size of 2 × 2, a batch normalization layer, a flatten layer to transform the output from the previous layer, a dense layer with 48 nodes, and a dropout layer with a rate of 0.5. The final output layer has 4 nodes (number of classes) and a softmax activation function. Finally, the model is compiled with categorical cross-entropy loss, Adam optimizer with a learning rate of 0.001, and the metric set to *accuracy*. The total parameters for VGG16 are 14,714,688; 42,658,176 parameters for ResNet V2; and 8,769,374 parameters for EfficientNet.

This solution designs an architecture of Deep Convolutional Neural Networks. This architecture includes convolutional layers in six blocks, as well as some batch normalization layers. In this model, also called DS-CNN, each convolutional block has one CNN 2D layer, followed by a batch normalization, an ELU activation, and a max pooling 2D layer. For this architecture, any dropout layers with a dropout ratio of 0.4 are defined. Figure 7 shows the architecture of DS-CNN.

In this way, this model is a sequential model with several convolutional blocks and fully connected layers. The model architecture is as follows: First and sixth CNN blocks consist of a convolutional layer with 256 filters; the second and fifth CNN blocks contain a convolutional layer with 192 filters; and third and fourth CNN blocks comprise a convolutional layer with 128 filters. All six blocks include a kernel size of (2,2) and a relu activation function. The flatten layer flattens the output from the previous layers. The first fully connected layer is a dense layer with 256 units and a relu activation function. The output layer is a dense layer with the number of classes and a softmax activation function. The model is compiled using the Adam optimizer with a learning rate of 0.001. The loss function is categorical cross-entropy, and the metric used for evaluation is accuracy. This architecture aims to classify data into multiple classes using convolutional layers followed by fully connected layers.

The DS-CNN model has a total of 151,653,764 parameters, including both trainable and nontrainable parameters. The trainable parameters are those that are updated during the training process to optimize the model’s performance. In this case, all 151,653,764 parameters are trainable, indicating that the model can learn and adjust the values of these parameters based on the training data. The nontrainable parameters, on the other hand, are fixed and do not change during training. These parameters are usually associated with pretrained layers or frozen layers in the network. In this model, there are no nontrainable parameters, implying that all parameters are adjustable and updated during the training process.

The outcomes of this stage of the project show that DS-CNN does not show overfitting and increases the efficiency regarding other architectures. Overall accuracy reaches 79% and, among the four emotions, surprise is the one that obtains the best results. Figure 8 shows the main result of DS-CNN. Figure 8a,b, along with Table 2, depict, respectively, the accuracy, confusion matrix, and the classification report of the model. The model can be observed to have high percentages of assertiveness and accuracy, with an average of 79%. This indicates that it is a robust model that has significant opportunities for improvement. We can determine if there is no overfitting in a deep learning model by analyzing its performance on both the training and validation datasets. Overfitting occurs when a model performs exceptionally well on the training data but fails to generalize to new, unseen data (validation or test data). If the training and validation accuracies are both high and show similar trends as the model is trained, this points out that the model is not overfitting. To further confirm the absence of overfitting, we can compare the model’s accuracy on the training and validation datasets throughout the training process. If the training accuracy continues to increase while the validation accuracy plateaus or remains stable, it suggests that the model is not memorizing the training data but rather learning to generalize to new data. Running the model for more epochs may lead to better results if there is no overfitting, because the model continues to learn from the training data and improve its performance. However, if overfitting is present, continuing to train the model for more epochs can worsen its generalization capabilities, as it will increasingly memorize the training data and lose the ability to generalize to new data. Finally, by monitoring the training and validation accuracies and ensuring they both perform well without significant divergence, we can conclude that there is no overfitting. In such cases, running the model for more epochs may lead to improved results and better generalization.

The decision to stop early and determine the appropriate number of epochs in Figure 8 is based on monitoring the model’s performance by using the validation data. Overfitting occurs when a model learns to perform exceptionally well on the training data; however, it fails to generalize unseen data, such as the validation set. To address this concern, we continuously tracked the model’s performance on the validation data during training. If we noticed that the validation performance started to degrade or stagnate while the training performance continued to improve, it indicated a sign of overfitting. In Figure 8, the decision to stop at epoch 30 is a result of this monitoring process. We observed that the validation performance reached a plateau after epoch 30, suggesting that the model’s ability to generalize was stabilized. Thus, stopping at this point helps prevent overfitting and ensures that the model achieves the best possible performance on unseen data.

### 3.3. Training of the Convolutional Neural Network through Electroencephalogram

Utilizing EEG signals in a CNN offers promising advantages in the field of brain–computer interfaces (BCIs). EEG provides direct access to the brain’s electrical activity, allowing real-time monitoring and an analysis of cognitive states. By integrating EEG signals into a CNN architecture, we can leverage the temporal and spatial information encoded in the brain signals to enhance the learning and decision-making capabilities of the network. This procedure complements this setup by enabling the network to adapt and optimize its behavior based on feedback and rewards received from the environment. This combination holds the potential to unlock new possibilities in the development of advanced neurotechnologies, such as brain-controlled prosthetics, cognitive enhancement systems, and personalized healthcare applications. Overall, the incorporation of EEG in a CNN framework offers a powerful approach to harness the brain’s capabilities, paving the way for innovative advancements in brain–computer interfaces and human–machine interaction.

In this way, an EEG signal is the measurement of the electricity flowing during the synaptic excitation of dendrites in the pyramidal neurons of the cerebral cortex. When neurons are activated, electricity is produced inside dendrites, generating a magnetic and electric field that is measured with EEG systems. The human head contains different layers (the brain, the skull, and the scalp) for attenuating the signal and adding external and internal noise; thus, only a large group of active neurons can emit a potential to be measured or recorded using surface electrodes [20]. The electrical activity of the brain can be captured on the scalp, at the base of the skull with the brain exposed, or at deep brain locations. The electrodes that acquire the signal can be superficial, basal, or surgical. For EEG, superficial electrodes are used [21].

Generally, for the acquisition and recording of the brain’s electrical activity in BCI systems, surface electrodes are used on the scalp. There are several ways to accommodate them, called acquisition systems, or simply arrays, e.g., Illionis, Montreal Aird, Lennox, etc., but the most used for research purposes, and certainly in this study, is the International 10-20 Positioning System [22].

The 10-20 International System is a standardized protocol based on the anatomical references of the inion and nasion longitudinally and the earlobes transversely, ensuring that electrodes are placed on the same areas, regardless of head size. Also, this system is named after its use of 10% or 20% of the anatomically specified distances to place the electrodes, as shown in Figure 9, from the nasion to the inion and from the earlobes [23], pointing out that they are the center.

This work uses the EEG device ThinkGear TGAM1, since it collects neural signals and inputs them into the ThinkGear chip for processing the signal into a sequence of usable data and filtering any interference in a digital way. Raw brain signals are amplified and processed to provide concise contributions to the device. The device considered to be the interface between the brain and the computer is the NeuroSky ThinkGear module, shown in Figure 10, since it meets the requirement of being a noninvasive method, while also having a reliability rate of 98%. The electrode is placed in position $F_{p1}$ [24,25].

The electroencephalographic sample database consists of 124 time-varying samples containing between nine thousand and fifteen thousand records each, with a range between 18 and 30 s. Table 3 shows a sample of the first 10 brain activity values from both raw signals and preprocessed signals, such as Delta, Theta, Low Alpha, High Alpha, Low Beta, High Beta, Low Gamma, and High Gamma frequency bands (Figure 11), along with the subject’s percentage of attention (Atte) and meditation (Med) during the test. The EEG sensor TGAM1 sampling rate is 512 samples per second for only the *RAW* time series, while the rest of the time series it is one sample per second. It is important to mention that all brain activity rhythms, attention and meditation, are preprocessed within TGAM1 at a rate of 1 Hz, and their equations are proprietary to NeuroSky. Therefore, based on Equation (Equation 1), the RAW signal is converted to microvolts (μV) and the power spectral density is calculated. (1)$$Volts=RAWdata\times \frac{\frac{1.8}{4096}}{2000}\;\left[\mu V\right].$$


According to the above, these signals of brain activity are included in the convolutional neural network model, described in Figure 7, resulting in the model shown in Figure 12. After testing transfer learning, fine tuning, EEG reinforcement, ensemble model, and recreating DCNN from scratch (even with 1 layer input—greyscale), it was found that fine-tuning can improve the model’s performance. Thirty models were built to test these methods, and the best-performing model was the DCNN with fine-tuning + dense blocks, achieving a testing accuracy of 80%. The model was trained for 30 epochs, using a batch size of 32, the Adam optimizer with a learning rate of 0.001, and the categorical cross-entropy loss function. The structure of the model consisted of two main parts: (i) DCNN with an input shape of 48×48×3, with 7 first layers frozen and the remaining layers trainable, and (ii) flattening, with three dense blocks with dense layers of 1024 neurons and batch normalization.

It is important to note that there is a very high probability that there is a desynchronization in both datasets, but they could still be used together by aligning their timestamps and ensuring that the data correspond to the same events or time intervals. To use the image and EEG datasets together, the following six steps were taken:Preprocessing: Preprocess both image and EEG data separately to ensure they are in a compatible format and aligned with the same timestamps or time intervals.Timestamp Alignment: The data were recorded simultaneously. Despite this, an alignment was made with the timestamps, or the synchronized the datasets were matched to the corresponding events or time periods. This can be achieved through careful data preprocessing and timestamping.Feature Extraction: Extract relevant features from both datasets, such as facial expressions from images and specific EEG features related to emotions.Fusion: Combine the features extracted from both datasets to create a unified feature representation that includes both visual and EEG information. This fusion step could involve concatenating the features or using more sophisticated techniques, such as multimodal learning.Model Training: Train a deep learning model, such as a CNN, on the fused feature representation to recognize facial emotions using the combined information from both image and EEG data.Evaluation: Evaluate the model’s performance on a validation or test dataset to assess how well it can predict facial emotions based on the joint information from images and EEG.

In summary, it was possible to utilize both datasets together to enhance the model’s understanding of facial emotions and potentially improve the accuracy of the facial emotion recognition task.

## 4. Experiments and Results

### 4.1. Down Syndrome Dataset Findings

First, DS-CNN is trained, tested, and validated without EEG signal along the Down Syndrome Dataset, obtaining the following results. Thus, when DS-CNN is applied to the Down Syndrome Dataset, results do not show overfitting. Overall accuracy reached 72%, and surprise is barely recognized in any case, but the other emotions are recognized almost perfectly. Figure 13 shows the main result of DS-CNN. Figure 13a,b, in addition to Table 4, depict, respectively, the accuracy, confusion matrix, and classification report of the model.

This project is promising in terms of the results and their application. In future implementations, it would be important to use more robust architectures of Deep Convolutional Neural Networks that improve the results. Moreover, for implementation, it is necessary to increase the number of images in the DSDS and to establish a solution to capture the emotion of surprise, because it is normally confused with calm. With these modifications, it will be possible to measure brain activity in children with Down Syndrome when they are calm and there are no disturbances caused by some secondary emotion.

### 4.2. Discussion

It is noteworthy that multifactorial ANOVA is a statistical technique used to analyze the effects of two or more independent variables (factors) on a dependent variable. In the context of this study, it could be used to analyze how different factors, such as image features, EEG data, and possibly other variables, influence facial emotion recognition accuracy. To perform multifactorial ANOVA on our limited dataset, the following six steps were taken:Factor Identification: Determine the independent variables (factors) to be included in the analysis. In this case, it could involve factors related to image features, EEG features, and other relevant variables, such as overtraining or overfitting, facial angle, head accessories, and the recognizer.Data Preparation: We prepare the data for analysis, ensuring that it is organized into appropriate groups based on the different levels of each factor. For example, images and EEG data were grouped based on different emotion categories.Model Specification: Set up the ANOVA model, specifying the dependent variable (efficiency of facial emotion recognition accuracy) and the factors that are being investigated. The model will assess how the different factors contribute to the variation in the dependent variable.Hypothesis Testing: Conduct hypothesis testing to determine whether there are significant effects from the different factors on facial emotion recognition accuracy. This involves calculating F-statistics and *p*-values for each factor.Post Hoc Tests: If significant effects are found, post hoc tests may be conducted to determine which specific levels of the factors are significantly different from each other.Effect Size: Calculate effect sizes to assess the practical significance of any significant effects found.

Now, for testing the main factors that affect this proposal, a multifactorial ANOVA over the F1 Score is performed, which is a procedure that performs an analysis of variance of several factors for *F1 Scores*. In addition, various tests and graphs are performed to determine which factors have a statistically significant effect on the *F1-Score*. Any ANOVA table contributes to the identification of significant factors. For each significant factor, the *Multiple Range Tests* distinguish which means are significantly different from others. For the factors *overtraining* and *recognizer*, there are no statistically significant differences between any pair of means, with a confidence level of 95. 0%. For the variables *accessory* and *facial angle*, they indicate that there are statistically significant differences with a confidence level of 95.0%. The analysis of variance for F1-Score—Type III Sum of Squares, Table 5, decomposes the variability of the *F1-Score* metric into contributions due to various factors. Since Type III sum of squares (default in Statgraphics) was chosen, the contribution of each factor is measured by removing the effects of the other factors. The *p-values* test the statistical significance of each of the factors. Since the *p-values* for *B. accessory* and *C. facial angle* are less than 0.05, these factors have a statistically significant effect on *F1-Score* at a 95.0% confidence level; see Table 5.

The method currently used to discriminate between the means is the *Fisher’s Least Significant Difference (LSD)* procedure, shown in Figure 14. With this method, there is a 5.0% risk of saying that each pair of means is significantly different when the real difference is equal to 0. To interpret Fisher’s *LSD*, the distance between the extremes of the means is examined. The smaller the distance between the points, the smaller the significance of the factor. The analysis of variance over F1 Score—Sum of Squares Type III, decomposes the variability over the *F1 Score* metric into contributions due to various factors. Since the Type III sum of squares was chosen, the contribution of each factor is measured by removing the effects of the other factors. The *p-values* test the statistical significance of each of the factors. Since the *p-values* for *B. accessory* and *C. facial angle* are less than 0.05, these factors have a statistically significant effect on the *F1-Score* with a 95.0% confidence level. Thus, based on the results obtained, it is concluded that DS-CNN performs better when the patient faces the camera and the measurement accessories that are placed are more discreet.

Once the appropriate Deep Convolutional Neural Network model is established and all factors that do not influence the experiment are eliminated, the EEG signals are included in the model training. Thus, Figure 15 shows that the model overfits from epoch 37 onward, which is why only 30 epochs are chosen for its final training. The usage of the transfer learning technique proved to enhance the models’ performance. However, the base models were trained using the Down Syndrome Dataset, which differs significantly from the Exploratory Data Analysis database utilized in this project. Therefore, many layers of these base models had to be retrained, which posed a challenge to the advantage of utilizing transfer learning techniques—shorter training times. Although this model exhibited high precision (low false positives) and recall (low false negatives) when classifying *attention* and *surprise* faces, its biggest challenge is still in distinguishing *dislike* and *calm* expressions, as evidenced by the lower performance of these emotions, as shown in Table 6.

Also, Table 6 shows an average efficiency of 81%, which increases the results by 11% when brain activity is included in the training and validation of the model in Figure 12.

Finally, to test the efficiency of our model, we employ the area under the curve (AUC), which is a metric used in classification problems to evaluate the performance of a model. The AUC’s values range from 0 to 1, where a value of 1 indicates that the model is capable of perfect classification, while a value of 0 manifests its disability to classify correctly; the exact value of the AUC that is considered good or acceptable may vary depending on the problem and context in which the model is applied. In general, an AUC value greater than 0.5 indicates that the model is better than chance in classification, while an AUC value close to 1 indicates good classification performance. In this case, we obtained an AUC value of 0.8269, pointing out that the model has good classification ability and is capable of effectively distinguishing between facial emotions with respect to brain activity.

## 5. Conclusions

Several works related to the analysis of artificial vision, emotional responses, and Trisomy 21 were discussed. These works were selected to use artificial vision to analyze nonverbal emotional language, and one of them even analyzes the neural activity of children with Trisomy 21. The main differences are the body part that is valued to determine an emotion and the target audience. Our results show an average efficiency of 81%, which increases the results by 9% when brain activity is included in the training and validation of the proposed DCNN model. We were also able to prove that EEG reinforces CNN by providing additional brain activity data that complements the visual data, enabling the model to gain deeper insights into the emotional expressions and improving its ability to recognize and understand facial emotions.

To test the functioning of DS-CNN, a series of 16 experiments were designed with interaction between four variables: *(i) recognizer, (ii) accessory, (iii) facial angle, and (iv) overtraining*; and the experimentation was repeated with two patients. The values generated for *F1-Score* and their possible combinations were examined by means of a *multifactorial analysis of variance*. This analysis managed to highlight that using more discreet accessories for sampling and capturing the patient from the front contributes to better performance of the *F1 Score* metric.

In the present work, DS-CNN, a system focused on microexpression analysis in children with Trisomy 21 was designed, contributing to the study of the effectiveness of dolphin-assisted therapy (DAT) by indicating the emotion expressed on the face of children throughout the process. With the development of DS-CNN, the feasibility of the scenarios and the environments where it can be used during DAT was also evaluated, and with this, rethinking the way it will work together with other analysis systems.

We can list two main contributions of this work: (i) It is a novel approach, since the study introduces a new approach by combining deep learning with EEG data for facial emotion recognition; this integration allows for a more comprehensive analysis of emotional responses in children with Down Syndrome during a DAT. (ii) It allows the possibility of real-time detection, since the proposed model achieves real-time facial emotion recognition, which can have practical applications in monitoring and providing timely feedback during therapy sessions. However, some limitations should be considered in future releases, such as that this study uses a limited dataset for training the deep learning model. This could potentially limit the generalizability of the results and may not fully capture the variability in facial emotions of children with Down Syndrome during DAT, since the size of the testing dataset is relatively small compared with the validation dataset, which could affect the robustness of the model’s performance evaluation.

There are no other studies using this same dataset, making direct comparisons of the proposed approach with similar works challenging. Nonetheless, the absence of previous studies using this dataset does not diminish the significance of the study’s contributions. Instead, it highlights the novelty and originality of the proposed approach, which explores a unique combination of EEG data and deep learning for facial emotion recognition in children with Down Syndrome during DAT.

Finally, the future work of this project includes the following:Expand the target audience to include the analysis of microexpressions in people with Autism Spectrum Disorder, for instance.Improve the database by increasing it and having more active participation from the expert in microexpressions.Design a new series of experiments where variables such as the type of lighting and the type of sample to be analyzed are taken into account.Implement the analysis of microexpressions in real time.Introduce transfer learning to the neural network architecture.

## Figures and Tables

**Figure 1 healthcare-11-02295-f001:**
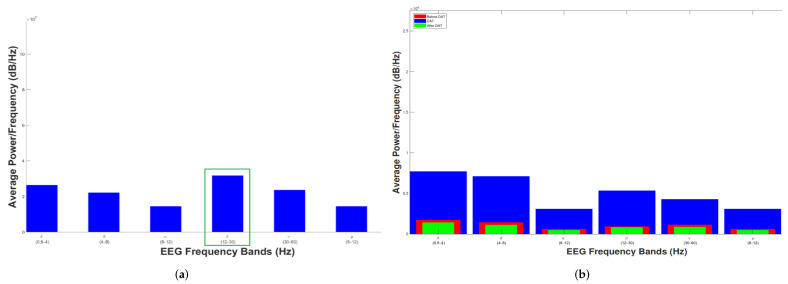
Dolphin and child brain activity during DAT: (**a**) Dolphin. (**b**) Children with disabilities.

**Figure 2 healthcare-11-02295-f002:**
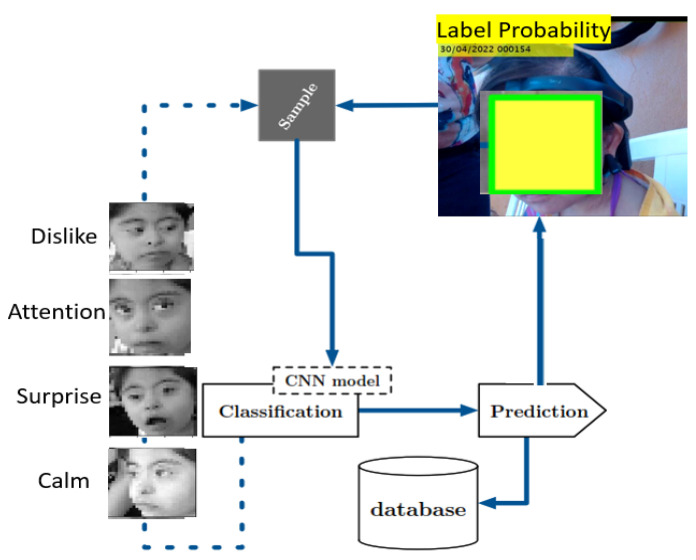
General Scheme of this proposal.

**Figure 3 healthcare-11-02295-f003:**
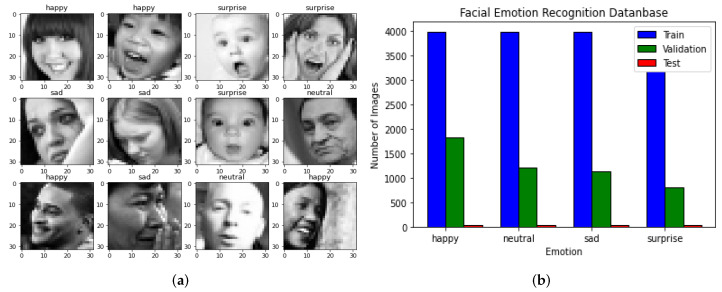
Reference datasets used in this project: (**a**) Image sample. (**b**) Size and distribution.

**Figure 4 healthcare-11-02295-f004:**
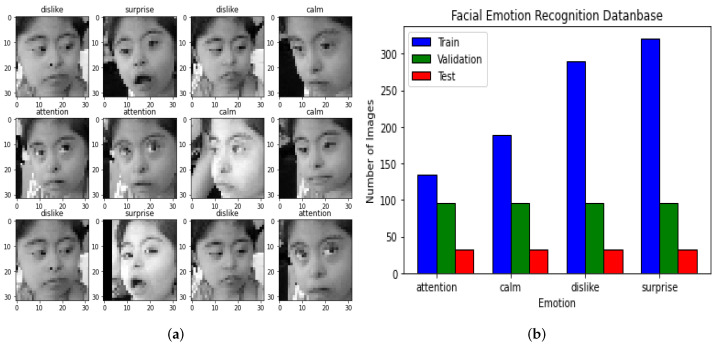
Down Syndrome Dataset included in this project: (**a**) Image sample. (**b**) Size and distribution.

**Figure 5 healthcare-11-02295-f005:**
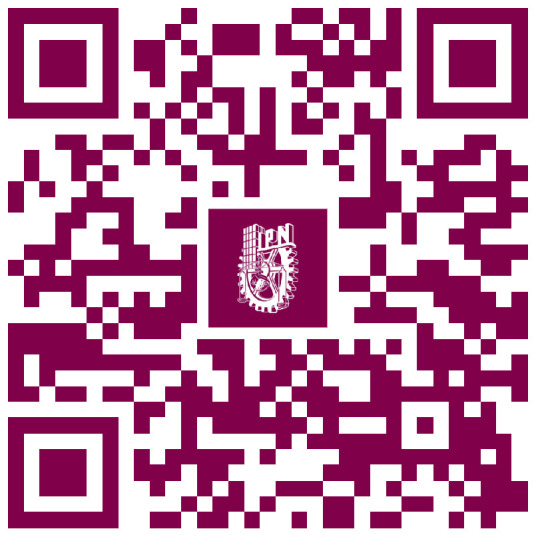
QR code in Google Drive where both EDA and DSDS can be downloaded.

**Figure 6 healthcare-11-02295-f006:**
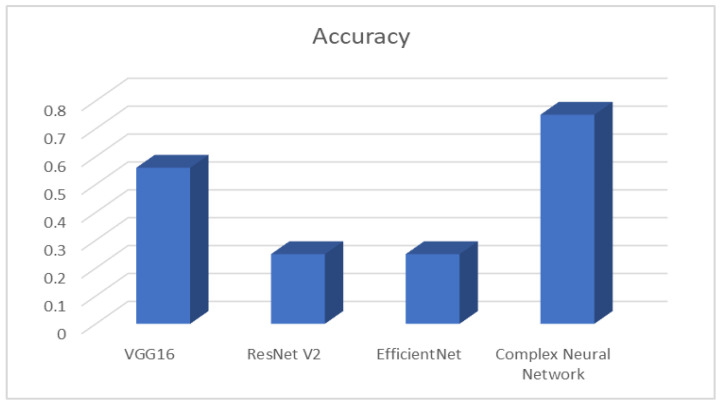
Comparison of various techniques and their relative performance.

**Figure 7 healthcare-11-02295-f007:**
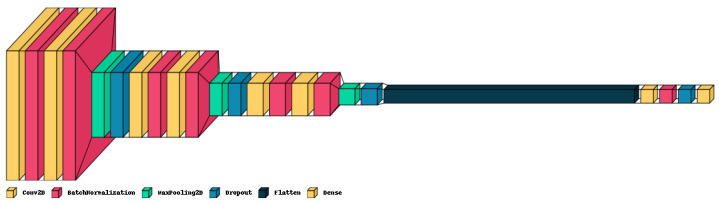
Deep Convolutional Neural Network architecture facial emotion scheme for DS-CNN.

**Figure 8 healthcare-11-02295-f008:**
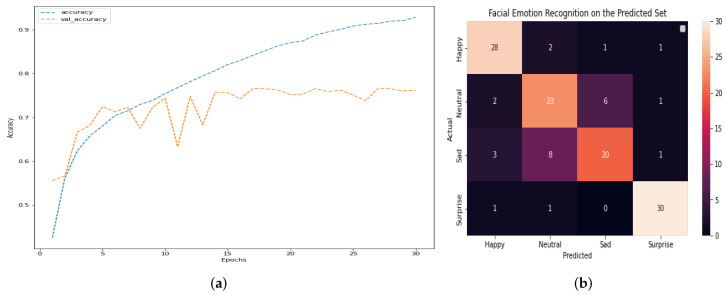
Main results of the Deep Convolutional Neural Network architecture DS-CNN: (**a**) Accuracy. (**b**) Confusion matrix.

**Figure 9 healthcare-11-02295-f009:**
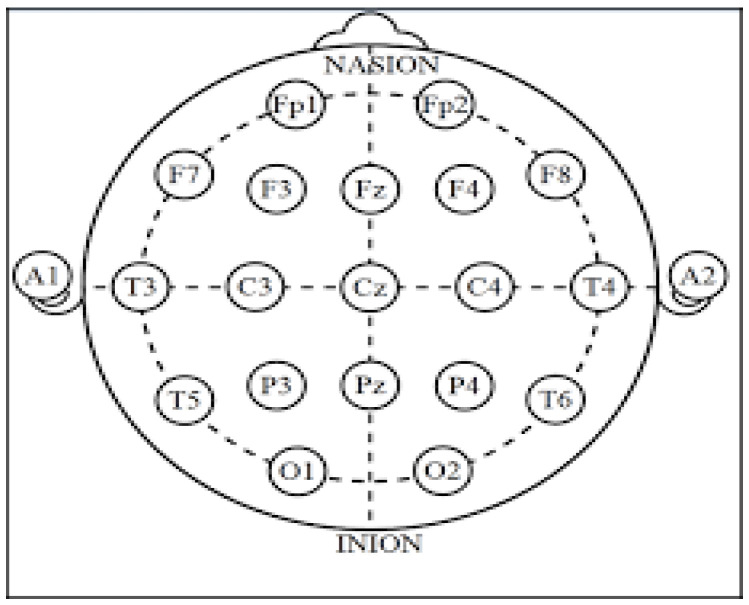
International Electrode Positioning System 10-20. In this work the $F_{p1}$ electrode is used.

**Figure 10 healthcare-11-02295-f010:**
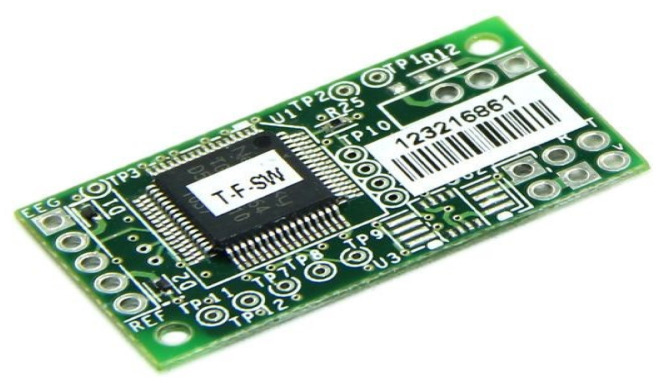
Sensor ThinkGear ASIC Module v1.0 (TGAM1).

**Figure 11 healthcare-11-02295-f011:**
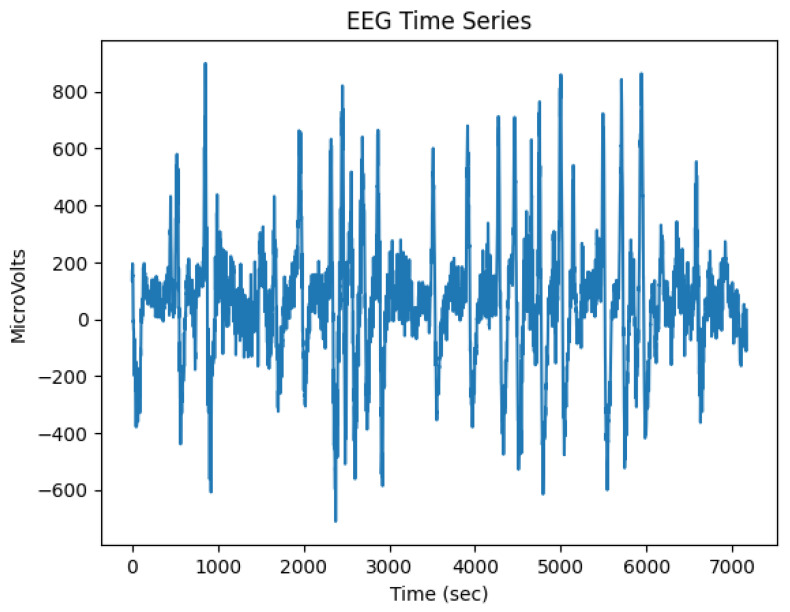
Example of brain activity when a panelist eats a functional product.

**Figure 12 healthcare-11-02295-f012:**
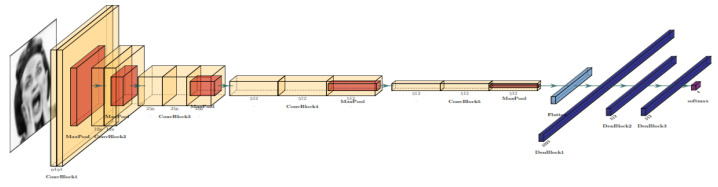
Deep Convolutional Neural Network Architecture: Facial Emotion step for DS-CNN.

**Figure 13 healthcare-11-02295-f013:**
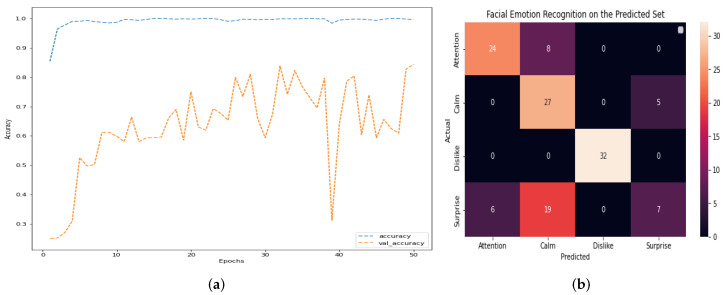
Main results of a Deep Convolutional Neural Network architecture: DS-CNN applied on Down Syndrome Dataset. (**a**) Accuracy. (**b**) Confusion matrix.

**Figure 14 healthcare-11-02295-f014:**
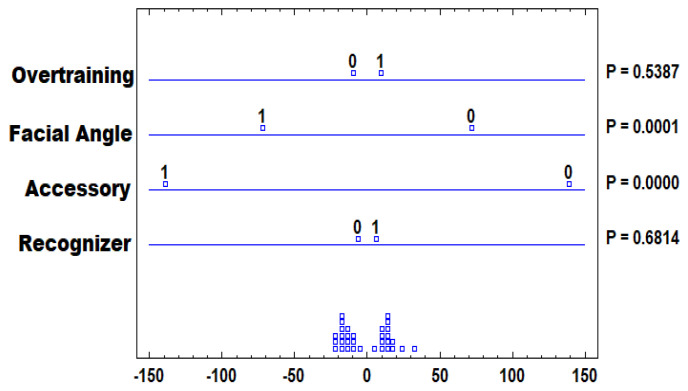
Fisher LSD for F1-Score of this proposal, *DS-CNN*, generated by means of *Statgraphics*.

**Figure 15 healthcare-11-02295-f015:**
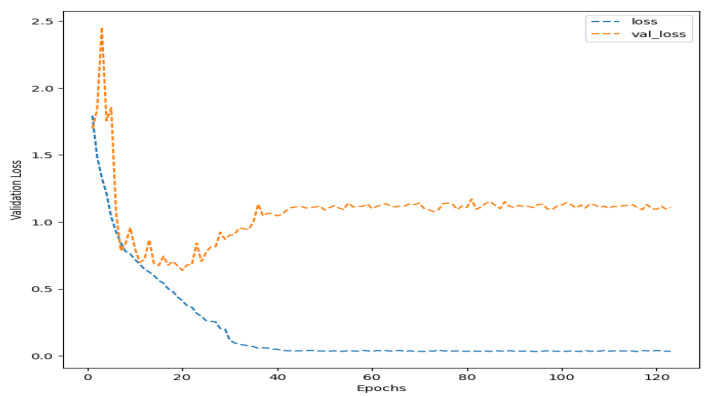
Fisher LSD for the F1-Score of this proposal, *DS-CNN*, generated by means of *Statgraphics*.

**Table 1 healthcare-11-02295-t001:** Comparison of various techniques and their relative performance.

Model	Happy	Neutral	Sad	Surprise	Accuracy
**VGG16**	0.53	0.48	0.5	0.74	0.56
**ResNet V2**	0	0	0	0.4	0.25
**EfficientNet**	0	0	0.4	0	0.25
**Proposed CNN Architecture**	* **0.81** *	* **0.61** *	* **0.7** *	* **0.9** *	* **0.75** *

**Table 2 healthcare-11-02295-t002:** Main Results of the Deep Convolutional Neural Network Architecture: Classification Report.

Emotion	Precision	Recall	F1-Score
**Happy**	0.82	0.88	0.85
**Neutral**	0.68	0.72	0.70
**Sad**	0.74	0.62	0.68
**Surprise**	0.91	0.94	0.92

**Table 3 healthcare-11-02295-t003:** Electroencephalographic sample of the brain activity of a children with Down Syndrome during a dolphin-assisted therapy.

RAW	Delta	Theta	Low Alpha	High Alpha	Low Beta	High Beta	Low Gamma	High Gamma	Attention	Meditation
−18	1771724	1435472	388994	229844	73905	61729	169719	67597	48	
−18	1771724	1435472	388994	229844	73905	61729	169719	67597	48	56
−38	1771724	1435472	388994	229844	73905	61729	169719	67597	48	56
−39	1771724	1435472	388994	229844	73905	61729	169719	67597	48	56
−41	1771724	1435472	388994	229844	73905	61729	169719	67597	48	56
−21	1771724	1435472	388994	229844	73905	61729	169719	67597	48	56
−18	1771724	1435472	388994	229844	73905	61729	169719	67597	48	56
−102	1771724	1435472	388994	229844	73905	61729	169719	67597	48	56
−214	1771724	1435472	388994	229844	73905	61729	169719	67597	48	56
−278	1771724	1435472	388994	229844	73905	61729	169719	67597	48	56
−253	1771724	1435472	388994	229844	73905	61729	169719	67597	48	56

**Table 4 healthcare-11-02295-t004:** Classification Report on the main Results of a Deep Convolutional Neural Network Architecture: DS-CNN Applied on Down Syndrome Dataset.

Emotion	Precision	Recall	F1-Score
**Attention**	0.80	0.75	0.77
**Calm**	0.50	0.84	0.63
**Dislike**	1.00	1.00	1.00
**Surprise**	0.58	0.22	0.32

**Table 5 healthcare-11-02295-t005:** Analysis of Variance for F1-Score—Type III Sum of Squares.

Source	Sum of Squares	df	Mean Square	F-Ratio	*p*-Value
Main Effects					
A: Recognizer	50	1	50	0.17	0.6814
B: Accessory	22,898	1	22,898	78.91	0
C: Facial Angle	6160.5	1	6160.5	21.23	0.0001
D: Overfitting	112.5	1	112.5	0.39	0.5387
Residuals	7834.88	27	290.181		
Total (Corrected)	37,055.9	31			

**Table 6 healthcare-11-02295-t006:** Classification Report the main Results of a Deep Convolutional Neural Network Architecture: DS-CNN applied on Down Syndrome Dataset using Brain Activity.

Emotion	Precision	Recall	F1-Score
**Attention**	0.96	0.84	0.90
**Calm**	0.77	0.72	0.74
**Dislike**	0.64	0.84	0.73
**Surprise**	0.93	0.81	0.87

## Data Availability

Not applicable.

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
