# Peer review of "DS-CNN: Deep Convolutional Neural Networks for Facial Emotion Detection in Children with Down Syndrome during Dolphin-Assisted Therapy"

_healthcare, 2023, doi:10.3390/healthcare11162295_

Round 1

Reviewer 1 Report (Previous Reviewer 1)

The paper at hand proposes a method for facial emotion recognition in children with Down Syndrome from static images. To achieve that a deep Convolutional Neural Network (CNN) architecture is exploited and trained to assess its performance. The efficiency of the classifier is evaluated on two databases named First Data Set (FDS) and Second or Down’s Syndrome Data Set (DSDS).

To begin with, the task of facial emotion recognition on children with Down syndrome is indeed a challenging field and can really enhance the contributions of artificial intelligence and computer vision in the healthcare domain. Yet, the paper fails to a great extent to provide a competitive and reliable solution.

The presentation of the paper is very poor and needs to be greatly improved.

Both sections 2 and 3 need to be completely reorganized. At this point, part of the methods is presented in both sections as well as the exploited datasets and the experimental study are partially described.

The introduction section fails to capture the field. In particular, more discussion should be conducted about typical emotion classification tasks, such as visual and multimodal emotion recognition. The authors discuss about versions of the YOLO detector and classification tasks from EEG that are both completely irrelevant to the introduced facial emotion recognition task. In my view, I highly recommend presenting the fields of visual emotion recognition and multimodal emotional recognition based on (a) handcrafted features, (b) geometric features (facial landmarks) and (c) appearance features (RGB images). Refer to the references below:

Kansizoglou, Ioannis, et al. "An active learning paradigm for online audio-visual emotion recognition." IEEE Transactions on Affective Computing 13.2 (2019): 756-768.

Chowdary, M. Kalpana, Tu N. Nguyen, and D. Jude Hemanth. "Deep learning-based facial emotion recognition for human–computer interaction applications." Neural Computing and Applications (2021): 1-18.

Ngoc, Quang Tran, Seunghyun Lee, and Byung Cheol Song. "Facial landmark-based emotion recognition via directed graph neural network." Electronics 9.5 (2020): 764.

Kansizoglou, Ioannis, et al. "Continuous Emotion Recognition for Long-Term Behavior Modeling through Recurrent Neural Networks." Technologies 10.3 (2022): 59.

Shehu, Harisu Abdullahi, Will Browne, and Hedwig Eisenbarth. "An adversarial attacks resistance-based approach to emotion recognition from images using facial landmarks." 2020 29th IEEE International Conference on Robot and Human Interactive Communication (RO-MAN). IEEE, 2020.

The exploited datasets are not sufficiently described. The authors should describe more how did they collect the data, how many subjects did they include, and how did they annotate, evaluate and split the dataset.

What is the clear architectural design of the proposed CNN (not the VGG and ResNet ones)? What is the number of layers, the kernel sizes and the number of hidden units? Figures 7 and 12 are not readable. The methodology section is not organized.

In Table 2, it is counter-intuitive that ResNet-V2 and EfficientNet architectures demonstrate such a poor performance, considering also that in general, they perform better than the VGG16 one. How did the authors train and evaluate those networks? How did they define their hyper-parameters? Extensive experiments on this subject are highly required.

The reason for displaying Table 3 is not clear. In general, the discussion regarding EEG signals is not relevant and out of the scope of the paper not mentioned in the title and the abstract, as well.

The number of experiments and the lack of sufficient descriptions fail to prove the novelty and efficiency of the proposed system.

Overall, the use of English needs to be greatly improved, by correcting typos, syntax and grammatical errors.

Author Response

Reviewer 2 Report (Previous Reviewer 3)

The authors used Deep Convolutional Neural Networks for Facial Emotion Detection in Children with Down Syndrome. The title includes facial emotion detection, whereas, in the paper, facial and brain activity are considered.

Introduction

1.     Last paragraph of page 1: How is the paragraph in green related to the therapies for children with Down Syndrome(DS)? This paragraph should be replaced with the different types of therapies used for the children of DS and how AI is used in these therapies.

2.     How does the knowledge of the emotions of DS children help in DAT therapy?

3.     The last paragraph of page 2 is not necessary.

4.     Page 2: The first objective is unclear and should be rewritten. How many methodologies are evaluated? What is the meaning of planning the proposal by contrasting related works? Other objectives are also vague. How theoretical understanding of concepts are done in the paper? What is the meaning of process and threads?

5.     Last paragraph of page 3: What is the difference between the goals mentioned here and the objectives listed above?

6.     Figure 1: What is the purpose of measuring dolphins' brain activity, and how is it done? Moreover, is it relevant to the study conducted in this paper?

7.     The title of section 2 should be the proposed methodology.

Section 2: Methods

8.     Figure 2 should be re-drawn and explained in the text. In the current form, it isn't easy to understand the flow of the figure.

9.     What is the meaning of reinforcement of facial expressions by including EEG?

10.  Exploratory data analysis (EDA) is used to analyze data sets and summarize their main characteristics. What is the meaning of EDA having three folders?

11.  Why is the testing dataset (128) small compared to the validation dataset (4977)? A smaller validation dataset is used for early stopping.

12.   What was the data distribution after data augmentation?

13.  Why is grayscale image better than color images? Any reference?

14.  A smaller dataset is not useful in training deep learning architectures from scratch. So whether you select any model based on a larger dataset is irrelevant. The best solution is using a larger dataset (DSDS) or the transfer learning concept.

15.  Table 1: A list of models appropriate for facial emotion recognition should be selected from the literature. In this table, ResNet and EfficientNet ultimately failed. So there is no need to show their results. Complex NN is not explained in the text that showed the best accuracy.

16.  What is the meaning of robustness accuracy?

17.  Figure 7 is not clear. What do the authors want to show in this figure?

18.  All four architectures in Table 1 are deep CNN, so how the authors selected deep CNN as their final solution. Based on the results in Table 1, how the authors decided on the architecture in Figure 7.

19.  Figure 8: How can we decide that there is no overfitting? If there is no overfitting, running for more epochs may give better results.

20.  Table 3 is not explained in the text. Is this raw data? What are the numbers in the table? How is the pre-processing done? Please describe the equations used.

21.  How EEG reinforces CNN?

22.  It is not clear how the data of images and EEG are recorded simultaneously. If not, then how can two datasets be used together?

23.  Figure 12: Explain all the stages of the model. How is EEG plugged in after DS-CNN?

Results

24.  The inclusion of more layers in DS-CNN increased the efficiency of other architectures…What are the other architectures, and how efficiency is measured? What is the meaning of 2% to 54%?

25.  What is the difference between the results in Tables 2 and 4? These are two different experiments or the same experiment. If different, then why low accuracy for the surprise?

26.  How is the multifactorial ANOVA done on this limited dataset? How are different factors analyzed? It needs more explanation and data description.

27.  How EEG and images are used together for a particular emotion?

Proofreading of the paper is required.

Author Response

Reviewer 3 Report (Previous Reviewer 4)

The authors did enough to address my previous comments. 

Author Response

Reviewer 4 Report (New Reviewer)

In the study, the researchers performed emotion analysis on the images of children with Down syndrome and applied to deep learning for this. From what I understand, the study appears to have undergone a previous revision and has been updated. However, removing the following issues will increase the readability and quality of the study.

1. In the study, a summary of the literature is given in the Introduction. Giving the literature summary under a separate title, “Literature Survey” or “Related Works” will increase the readability of the article.

2. The organization of the study should be given at the end of the Introduction. What has been described in other sections should be summarized here.

3. A figure (Figure 2) was immediately put under the Method section and no explanation was given. Before the figure, what happens in this part should be mentioned and Figure 2 should be explained.

4. On what basis is the data separated into training, testing and validation?

5. The training process was carried out with 20 epochs. An epoch value of 20 is a very low value. How can a successful training can be guaranteed with this epoch value? This should be discussed.

6. The quality of the figures is very low. Achieving higher resolutions will increase the legibility of the figures. As far as I know, MDPI journals require at least 300 DPI figures.

7. In the third section, the advantages and disadvantages of the study should be mentioned. What contributions have been made with this study and what limitations have been observed?

8. If there are studies using the same data set, it is necessary to compare these studies with the study. In this way, the success of the proposed approach can be seen clearly.

9. The AUC score is also frequently used in studies in the field of health. In the study, adding the AUC score to the results will increase the quality of the study. The AUC score should be included and interpreted.

In the study, the researchers performed emotion analysis on the images of children with Down syndrome and applied to deep learning for this. From what I understand, the study appears to have undergone a previous revision and has been updated. However, removing the following issues will increase the readability and quality of the study.

1. In the study, a summary of the literature is given in the Introduction. Giving the literature summary under a separate title, “Literature Survey” or “Related Works” will increase the readability of the article.

2. The organization of the study should be given at the end of the Introduction. What has been described in other sections should be summarized here.

3. A figure (Figure 2) was immediately put under the Method section and no explanation was given. Before the figure, what happens in this part should be mentioned and Figure 2 should be explained.

4. On what basis is the data separated into training, testing and validation?

5. The training process was carried out with 20 epochs. An epoch value of 20 is a very low value. How can a successful training can be guaranteed with this epoch value? This should be discussed.

6. The quality of the figures is very low. Achieving higher resolutions will increase the legibility of the figures. As far as I know, MDPI journals require at least 300 DPI figures.

7. In the third section, the advantages and disadvantages of the study should be mentioned. What contributions have been made with this study and what limitations have been observed?

8. If there are studies using the same data set, it is necessary to compare these studies with the study. In this way, the success of the proposed approach can be seen clearly.

9. The AUC score is also frequently used in studies in the field of health. In the study, adding the AUC score to the results will increase the quality of the study. The AUC score should be included and interpreted.

Round 2

Reviewer 1 Report (Previous Reviewer 1)

I acknowledge that the authors have made considerable efforts to improve the manuscript.

More specifically, the presentation of the introduced model has been expanded, as well as the description of the exploited datasets and the conducted experiments.

Yet, the overall presentation and the writing quality still need improvements. I strongly encourage the authors to give the manuscript to be processed by a native speaker.

The description of the architectural design in the last paragraphs of Section 3.1 should be enclosed in a unified Table instead of a dense paragraph.

The term "reinforcement learning" is not valid. To my understanding, the authors have exploited EEG signals to fuse the facial image features and enhance overall recognition accuracy. Reinforcement learning constitutes a specific field of machine learning, including an intelligent agent with specific state and action spaces, a policy or utility function and a reward function. The above is irrelevant to the proposed methodology by the authors. Please, remove/replace the term throughout the manuscript.

I strongly encourage the authors to give the manuscript to be processed by a native speaker.

Just some indicative English mistakes:

Page 6: "Also, EDA consists in three subsets"

Page 8: "VGG16, ResNet V2, and EfficientNet neural network models has the following architecture"

Author Response

Reviewer 2 Report (Previous Reviewer 3)

The authors revised the paper but still there are many points which are not addressed properly in the revised version. My new comments are in red color.

Figure 1: What is the purpose of measuring dolphins’ brain activity, and how is it done? Moreover, is it relevant to the study conducted in this paper?

Still, it is unclear what benefits we will get by measuring EEG signal of the Dolphin and how it relates to the DAT therapy of the children.  Who are getting benefits we it is beneficial to measure their EEG. By one example of EEG features of the Dolphin, show what type of information you will be getting and how this information will be used?

What is the meaning of reinforcement of facial expressions by including EEG? The reinforcement of facial expressions by including EEG refers to the process of enhancing the analysis and understanding of facial expressions by simultaneously recording and analyzing the corresponding brain activity using electroencephalography (EEG) techniques.

This is not reinforcement. You are concatenating two types of features from two different sensors to improve the accuracy. It will be interesting to see the accuracy of the emotion recognition by EEG only. What percentage of accuracy is improved by facial based recognition?

. Exploratory data analysis (EDA) is used to analyze data sets and summarize their main characteristics. What is the meaning of EDA having three folders? To comply with this observation, we added the following paragraph: The use of Exploratory Data Analysis (EDA) with three folders…

The word three folders is unnecessary.

Why is the testing dataset (128) small compared to the validation dataset (4977)? A smaller validation dataset is used for early stopping.

To comply with this observation, we added the following paragraph: The testing dataset (128) is smaller compared to the validation dataset (4977) because the testing dataset serves a different purpose in the model evaluation process. The validation dataset is used for fine-tuning the deep learning neural network and making adjustments to optimize its performance during the training process.

This explanation is not correct. Training dataset is used to optimize the deep NN. Validation set is only used for early stopping. So, the training set should be as big as possible, validation dataset is small, and testing dataset should be enough to test the trained network thoroughly.

What was the data distribution after dataaugmentation? To comply with this observation, we added the following paragraph: In this case, the original training dataset likely had 128 samples, and after applying data augmentation techniques, the dataset expanded to 4977 samples. The augmented data distribution now contains a more diverse set of examples, capturing different variations and perspectives of the input data.

So is it validation set of 4977 after data augmentation or training dataset of 4977 data points. The explanation is very confusing. Clearly explain the three datasets before augmentation and after augmentation. Data augmentation doesn’t capture the diversity of the input space. It only solve the problem of data imbalance to some extent.

A smaller dataset is not useful in training deep learning architectures from scratch. So whether you select any model based on a larger dataset is irrelevant. The best solution is using a larger dataset (DSDS) or the transfer learning concept. Indeed, when training deep learning architectures from scratch, a smaller dataset may not be sufficient to achieve good model performance due to the risk of overfitting. Overfitting occurs when the model memorizes the limited training data, leading to poor generalization to unseen data.

My question was not to propose solutions. Identify clearly in your paper, what technique you have used and how? If transfer learning is used, then what is the base trained model and how you performed the transfer earning?

Table 1: A list of models appropriate for facial emotion recognition should be selected from the literature. In this table, ResNet and EfficientNet ultimately failed. So there is no need to show their results. Complex NN is not explained in the text that showed the best accuracy.

To comply with this observation, we added the following paragraph: Table 1 presents a list of models suitable for facial emotion recognition, selected from the existing literature. However, during the experimentation process, both ResNet and EfficientNet models were found to be unsuccessful in achieving satisfactory results, and therefore, their performance is not included in the table. On the other hand, the text does not provide a detailed explanation of the Complex NN model that demonstrated the best accuracy among the listed models. Therefore, it is essential to provide further information and clarity regarding the architecture and functioning of the Complex NN model to better understand its superior performance in facial emotion recognition. The lack of an explicit explanation of the Complex NN model in the text may hinder the reproducibility and understanding of the research findings. As a result, a comprehensive description and analysis of the Complex NN model’s design and mechanisms should be added to the paper to address this limitation.

Where is the ANSWER? Tale 1 should be updated. Details of ResNet and EfficientNet should be removed from paper and table 1. Details of Complex NN should be added to the paper.

What is the meaning of robustness accuracy? To comply with this observation, we added the following paragraph: In this case we employ Robustness accuracy for refering to the ability of a model or system to maintain high accuracy and performance even when facing challenges, variations, or perturbations in the data or input conditions.

Robustness and Accuracy are two different terms. You can say that accuracy should be robust enough under different conditions. If you are studying robustness that show how accuracy was preserved under different conditions?

All four architectures in Table 1 are deep CNN, so how the authors selected deep CNN as their final solution. Based on the results in Table 1, how the authors decided on the architecture in Figure 7.

That was decided based on the best result of all the neural networks compared.

You mean Figure 7 shows the architecture of Complex NN?

Figure 8: How can we decide that there is no overfitting? If there is no overfitting, running for more epochs may give better results. To comply with this observation, we added the following paragraph:

The Paragraph may explain the concept of overfitting. But the question remains the same. How have you decided the early stopping in figure 8 using validation data? Why stop at epoch 30?

Table 3 is not explained in the text. Is this raw data? What are the numbers in the table? How is the pre-processing done? Please describe the equations used. To comply with this observation, we added the following paragraph:

Where are the equations of preprocessing steps?

Figure 12: Explain all the stages of the model. How is EEG plugged in after DS-CNN?

Thanks for the observation, Figure 12 only shows the architecture of the proposed neural network.

Please add one more figure to show all the stages and how EEG plugged in after DS-CNN.

The inclusion of more layers in DS-CNN increased the efficiency of other architectures. . .What are the other architectures, and how efficiency is measured? What is the meaning of 2% to 54%?

The reason is that other neural network models have accuracies as low as 25% but others are barely better than 2%.

Explanation is unclear.

What is the difference between the results in Tables 2 and 4? These are two different experiments or the same experiment. If different, then why low accuracy for the surprise?

Table 2 is the CNN model applied to a set of facial recognition images, while Table 4 is applied to a set of images taken of a girl with Down syndrome.

So, Table 2 is not required as the paper is focusing on Down Syndrome.

How is the multifactorial ANOVA done on this limited dataset? How are different factors analyzed? It needs more explanation and data description.

In response to this observation, we add the following paragraphs to the beginning of Section 3.2.

Steps are defined but how these steps are used in this particular case? Is there only one factor, f1-score or more factors? Please explain the whole process more clearly. What are the results of ANOVA? Show in the form of table.

In addition, various tests and graphs are performed to determine which factors have a statistically significant effect on the F1 Score. An ANOVA table contributes to the identification of significant factors.

Where is the table?

AUC value of 0.826904296875

Too many digits after the decimal point? Is it justifiable?

Proofreading by a native speaker is required.

Author Response

Reviewer 4 Report (New Reviewer)

The authors made all the requested revisions and gave their reasons. Therefore, the publication can be accepted as it is.

The authors made all the requested revisions and gave their reasons. Therefore, the publication can be accepted as it is.

Author Response

This manuscript is a resubmission of an earlier submission. The following is a list of the peer review reports and author responses from that submission.

Round 1

Reviewer 1 Report

The paper at hand proposes a method for facial emotion recognition in children with Down Syndrome from static images. To achieve that a deep Convolutional Neural Network (CNN) architecture is exploited and trained to assess its performance. The efficiency of the classifier is on two databases named First Data Set (FDS) and Second or Down’s Syndrome Data Set (DSDS). The task of facial emotion recognition on children with Down syndrome is indeed a challenging field and can really enhance the contributions of artificial intelligence and computer vision in the healthcare domain. Yet, the paper fails to a great extent to provide a competitive and reliable solution.

The presentation of the paper is very poor and needs to be greatly improved. The introduction and related literature sections fail to capture the field. In the introduction, more discussion should be conducted about typical emotion classification tasks, such as visual and multimodal emotion recognition. The same applies to the related literature as well, which needs to be reorganized. To my view, I believe that the fields of visual emotion recognition and multimodal emotional recognition shall be discussed by exploiting (a) handcrafted features, (b) geometric features (facial landmarks) and (c) appearance features (RGB images). Refer to references on the above topics.

The exploited datasets are not sufficiently described. Are they publicly available? If so please cite the corresponding papers. If not, the authors should describe how did they collect the data, how many subjects did they include, how did they annotate, evaluate and split the dataset and if they provide them for use.

No clear architectural design or figures describing the framework of the proposed CNN is provided. The methodology section is too short. What is the number of layers, the kernel sizes and the number of hidden units?

In Table 1, it is counter-intuitive that ResNet-V2 and EfficientNet architectures demonstrate such a poor performance, considering also that in general, they perform better than the VGG16 one. How did the authors train and evaluate those networks? How did they define their hyper-parameters? Extensive experiments on this subject are highly required.

The experimental setup of the proposed network is also missing. The authors have to define and experimentally show the selection of the network’s hyper-parameters, such as learning rate, batch size, number of epochs, optimizer, etc.

The number of experiments and the lack of sufficient descriptions fail to prove the novelty and efficiency of the proposed system.

Overall, the use of English needs to be greatly improved, by correcting typos, syntax and grammatical errors.

Several long parts in the Introduction and Section 2 are exactly the same! The authors have to avoid writing the same sentences twice in the manuscript. E.g. “Evaluate the methodologies focused on analyzing emotional response for the planning of the proposal by contrasting related works.”

In the 3rd paragraph of the Introduction, "TAD" => "DAT". Search the entire manuscript and correct this typo.

In the 4th paragraph of the Introduction, please prefer "computer/machine vision" instead of "artificial vision"

"On the other hand, analyzing Facial Expressions of children with Down’s Syndrome to identify emotions during an

Assisted Therapy with Dolphins through Artificial Vision by means of a Deep Convolutional Neural Network." Please rephrase. The main verb is missing.

Reviewer 2 Report

Dear authors,

The report is good enough. However, a few changes have to be made.

1. Abstract: Good.

2. Introduction: It is good. However, deep learning part must be explained more. Deep learning algorithms are used to diagnose various diseases. The following articles are recommended to be added.

1. Krishnadas P, Chadaga K, Sampathila N, Rao S, Prabhu S. Classification of Malaria Using Object Detection Models. InInformatics 2022 Sep 27 (Vol. 9, No. 4, p. 76). MDPI.

2. Sampathila N, Chadaga K, Goswami N, Chadaga RP, Pandya M, Prabhu S, Bairy MG, Katta SS, Bhat D, Upadya SP. Customized Deep Learning Classifier for Detection of Acute Lymphoblastic Leukemia Using Blood Smear Images. InHealthcare 2022 Sep 20 (Vol. 10, No. 10, p. 1812). MDPI.

3. Chadaga K, Prabhu S, Sampathila N, Nireshwalya S, Katta SS, Tan RS, Acharya UR. Application of Artificial Intelligence Techniques for Monkeypox: A Systematic Review. Diagnostics. 2023 Feb 21;13(5):824.

4.Chadaga K, Prabhu S, Bhat V, Sampathila N, Umakanth S, Chadaga R. A Decision Support System for Diagnosis of COVID-19 from Non-COVID-19 Influenza-like Illness Using Explainable Artificial Intelligence. Bioengineering. 2023 Mar 31;10(4):439.

Please mention more details of the dataset. Was ethical clearance granted? Was it public data? Please mention these things.

You have added the accuracy curve for the DL algorithm, add the loss curve too. Further , classification report can be tabulated instead of copy pasting image.

Add limitations of the study along with future work. Paper is good enough, Please make the changes and resubmit. 

Reviewer 3 Report

The authors used Deep Convolutional Neural Networks for Facial Emotion Detection in Children with Down Syndrome. 

1.     Abstract: What is the large reference image dataset, and why this dataset is used?

2.     In the abstract, the authors should explain the proposed solution and its main features. Also, explain the outcomes, significance, and impact of the proposed solution.

Introduction

3.     Describes the background by illustrating the general area of research. It also mentions the importance of the selected research area by highlighting its critical factors. More references must be cited to explain different types of therapies.

4.     Briefly overview the existing practices and limitations of the existing practices and explain the research gap. How is emotion detection in children with Downs Syndrome different from standard facial emotion detection?

5.     How your proposed solution will solve the limitations?

6.     Page 2: The use of image processing and machine learning algorithms in emotion detection can be summarized with the help of a comparative table. Also, provide related references in the last three or four years.

7.     List of objectives mentioned at the end of the introduction section is not addressed in the paper.

Context

9.     What is the meaning of the context, and why it is included as a section in the paper?

10.  Why measuring the brain activity of dolphins is important and how is it related to the topic of the paper?

11.  Page 3: Lines 1 to 3: repetition of the text already discussed in the introduction.

12.  Authors have taken a list of objectives and goals from their project report and mentioned them here without emphasizing their role in the paper.

13.  Deep learning techniques in emotion detection may be explored in detail as a small literature review, as I mentioned above.

Method

14.  Describe the first dataset in detail. How many subjects are included, and what are their demographic details?

15.  Why testing dataset is very small?

16.  What is the purpose of using the first dataset FDS? Emotions in FDS and DSDS are different.

17.  How best architecture found in FDS will be helpful in DSDS?

18.  Table 1: the authors are using various deep learning architectures. There is no information in the paper about finding the best architecture or optimizing the architecture.

19.  Table 1: the results are very strange as the top three architectures completely failed to detect some of the emotions. Also, these four architectures are not defined in the paper and not appropriately cited.

20.  What is complex neural network architecture, and how is it used?

21.  Simple accuracy is not a good performance measure. The authors should have used precision, recall, and f1-score also.

22.  Figure 5: How this architecture differs from the complex neural network you selected. Or all four architectures are rejected, given in Table 1. Remember, three of the architectures in Table 1 are deep CNN.

23.  Describe in detail the DS-CNN used and show the learning mechanism of the parameters.

24.  Figure 6: Explain the purpose of figure (a). What is the meaning of the main results?

25.  Figure 7: How can a deep CNN architecture be trained with such a small dataset? Figure (a) shows strange behavior on validation data.

26.  Classification results must be compared with already published results.

27.  Page 7: last two paragraphs are very confusing, without any meaningful discussion.

Extensive editing of the English language is required. Sometimes, it is difficult to understand the sentence.

Reviewer 4 Report

- The abstract is too casual, it needs to be rewritten in a professional academic manner. For example, it is better to state the total number of children with Trisomy rather than annual births. Also, what is efficiency? It is not a standard metric for such problem.

- No need to capitalize  " Reference Image Database." or "Image Database"

- In the abstract, what does it mean to analyze their emotion?

- There is not introudction, section 1 starts with the related literature. This is the first line of your manuscript and should not start this way.

- No need to capitalize "Facial Expressions".

- The lack of a computer scientist is evident in the writing of the paper. The authors keep mentioning three folders; test, validate, and train, which makes it clear of their non-technical nature. 

- There is no proper description of the data. 

- There is no proper evaluation. What did you employ? K-fold, holdout?

- The evaluation metrics are not defined. 

- The samples in Figure 2a are no for Down's Syndrome sujects. 

- It would be great to look at the writing style and review the following related paper:

Fraiwan, M., Alafeef, M. & Almomani, F. Gauging human visual interest using multiscale entropy analysis of EEG signals. J Ambient Intell Human Comput 12, 2435–2447 (2021). https://doi.org/10.1007/s12652-020-02381-5

- typo Table 1, "Surprice"

- What is the utility of Fig. 5, which should have been described in the methods not the results. 

- The quality of some figures are very low.

- The table of abbreviations is missing but required by the journal template. 

Multiple typos.

Writing style.

Paper organization

Quality of figures 
